# Frequency, kinetics and determinants of viable SARS-CoV-2 in bioaerosols from ambulatory COVID-19 patients infected with the Beta, Delta or Omicron variants

S. Jaumdally[1,2], M. Tomasicchio[1,2], A. Pooran[1,2], A. Esmail[1,2], A. Kotze[1,2], S. Meier[1,2], L. Wilson[1,2], S. Oelofse[1,2], C. van der Merwe[1,2], A. Roomaney[1,2], M. Davids[1,2], T. Suliman[3], R. Joseph[4], T. Perumal [1,2], A. Scott [1,2], M. Shaw [3], W. Preiser [5], C. Williamson [4,6,7], A. Goga[8,9], E. Mayne [10,11,12], G. Gray[13], P. Moore [6,14,15], A. Sigal [16,17,18], J. Limberis [19], J. Metcalfe[19] & K. Dheda [1,2,20] ✉

Airborne transmission of SARS-CoV-2 aerosol remains contentious. Importantly, whether cough or breath-generated bioaerosols can harbor viable and replicating virus remains largely unclarified. We performed size-fractionated aerosol sampling (Andersen cascade impactor) and evaluated viral culturability in human cell lines (infectiousness), viral genetics, and host immunity in ambulatory participants with COVID-19. Sixty-one percent (27/44) and 50% (22/44) of participants emitted variant-specific culture-positive aerosols <10µm and <5µm, respectively, for up to 9 days after symptom onset. Aerosol culturability is significantly associated with lower neutralizing antibody titers, and suppression of transcriptomic pathways related to innate immunity and the humoral response. A nasopharyngeal Ct <17 rules-in ~40% of aerosol culture-positives and identifies those who are probably highly infectious. A parsimonious three transcript blood-based biosignature is highly predictive of infectious aerosol generation (PPV > 95%). There is considerable heterogeneity in potential infectiousness i.e., only 29% of participants were probably highly infectious (produced culture-positive aerosols <5µm at ~6 days after symptom onset). These data, which comprehensively confirm variant-specific culturable SARS-CoV-2 in aerosol, inform the targeting of transmission-related interventions and public health containment strategies emphasizing improved ventilation.

Over the past 3 years, coronavirus virus disease 2019 (COVID-19) has resulted in ~7 million recorded deaths worldwide[1] and has been one of the foremost infectious disease killers globally in the past three years. Transmission of SARS-CoV-2 probably occurs via the airborne route through the generation of a wide range of aerosol particles including large respiratory droplets (particle sizes up to 100 µm) that deposit with a fall time of several seconds to minutes primarily in the upper airways, and fine particles less than 10 µm with a suspension time of 30 min to several hours that deposit in the lower respiratory tract[2]. Historically, size-graded viable airborne viruses including influenza,

polio and vaccinia, have been isolated, using an Andersen cascade impactor system[3,4]. We have previously demonstrated similar capacity to isolate *M. tuberculosis* from size-graded aerosols sampled using an Andersen cascade impactor[5].

However, whilst airborne transmission of SARS-CoV-2 by respiratory droplets over short distances (1 to 2 metres) is well accepted, longer distance airborne transmission by fine aerosols (particles <10 μm) that can directly inoculate the deeper lung, remains contentious for several reasons including that it is difficult to demonstrate directly. Furthermore, although SARS-CoV-2 has been detected in hospital-based and community air samples, size fractionation and viral culture were not performed in most of these investigations[6–9]. Factors that support aerosol-based transmission have been cogently summarised[10] and include demonstration of long-range transmission in quarantine hotels[11], restaurants and choirs[12], demonstration of nosocomial transmission with masks that were permissive to aerosols but not droplets spread[13], identification of SARS-CoV-2 genomic material in building ducts and air conditioning systems, and in transmission ducted animal experiments[14,15]. However, many of these exemplars and phenomena remain anecdotal or inappropriate and could potentially be explained by prior or post-event exposure to COVID-19, sample contamination, technical factors, and may only be applicable to animal models or laboratory contexts. Although SARS-CoV-2 can remain viable for up to 3 h in laboratory generated aerosols, it has been challenging to isolate culturable or viable virus from human-generated aerosols (widely defined as <10 μm in median particle diameter and also for the purposes of this report), with only two studies demonstrating culturable virus in a total of five COVID-19 cases[16,17]. To our knowledge, there has been no systematic large scale study of culturability of virus in aerosols and whether this is consistent across variants or whether this is affected by proximity to symptom onset and with differing patterns of host immunity. Thus, critically, the final proof that human-generated aerosols <10 μm can harbour replicating virus remains largely unclarified.

To address this deficiency, we investigated the culturability of SARS-CoV-2 from size fractionated aerosols generated by ambulatory patients with COVID-19 during 2 follow-up visits (see Fig. 1 for a study overview). In summary, we found that between 25 and 60% of participants produced variant-specific culture positive aerosols <10 μm in diameter for up to 9 days after symptom onset, indicating considerable heterogeneity in the ability to produce culturable aerosols. Thus, only roughly a third of participants were probably highly infectious (produced culture-positive aerosols <5 μm at ~6 days after symptom onset) though we did not demonstrate person-to-person transmission. There was a relationship between nasopharyngeal PCR cycle threshold (Ct) value and aerosol culture positivity (including those probably highly infectious), and a 3 gene blood transcriptional signature was highly predictive of aerosol culture positivity. Although comorbidities and age did not predict aerosol culture positivity, the ability to produce culturable aerosols was associated with lower variant-specific neutralizing antibody titers and suppression of certain innate and adaptive components of host immunity (based on blood transcriptomic profiling).

## Results

### Study design and description of cases
Participants were recruited from a prospective observational cohort study investigating COVID-19 point of care diagnostic strategies in routine public and private health care settings in South Africa (IRB approval: University of Cape Town HREC 387-2020). Symptomatic ambulatory patients (≥18 years) with suspected COVID-19 and their asymptomatic contacts (≥18 years), presenting to selected health facilities or testing sites for investigation, were enrolled. Patients testing positive for COVID-19 on rapid antigen platforms or by PCR,

being within seven days of symptom onset and consenting to undergo cough aerosol sampling (CAS) and provide nasopharyngeal and saliva specimens were recruited. All included participants did not have severe disease (no need for hospitalization or supplemental oxygen). A total of 44 participants were enrolled and 38 of them completed a second visit for CAS within 2–3 days after the first procedure. There was an almost equal split across sex (21 males and 23 females), average age of participants was 38 years, ten (22%) reported a smoking history, 13 (30%) had received at least one dose of COVID-19 vaccine, and 11 (25%) reported having a risk factor other than a BMI > 25 (n = 22; 50%). Forty (91%) participants reported being symptomatic and 38 (86%) reported at least one respiratory symptom (cough, sore throat, shortness of breath or chest pain). Whole genome sequencing for viral variant classification (or date of infection) confirmed ten participants with Beta, 27 with Delta and 7 with Omicron. Table 1 provides a full breakdown of participants' characteristics.

### Frequency of aerosol culture positivity
First, we investigated the frequency and duration of culture positive aerosol generation. We found that between 50% (visit 2; a median of 6 days after symptom onset) to 60% (visit 1; a median of 4 days after symptom onset) of participants (n = 44) generated aerosols of <10 μm (0.65 to 7 μm particle median diameter), and between 30% (visit 2) to 50% (visit 1) produced aerosols <5 μm (0.65 to 4.7 μm) (see Fig. 2A that outlines the numbers and proportion of patients by visit). Thus, of the 44 participants, 13 were aerosol culture negative and 31 were aerosol culture positive (n = 15 at both visits; n = 7 at visit 1 but not visit 2; n = 4 at visit 2 but not visit 1; n = 5 at visit 1 but did not complete visit 2). Notably, we were able to culture virus from only 10% of viral medium impregnated settle plates that were located within 50 cm (0.5 m) from the patient. The complete size fragmentation of sampled aerosols (across 6 median particle sizes) for the two CAS visits is displayed in Fig. 2B.

### Heterogeneity of aerosol culture positivity
There was considerable heterogeneity in the ability of participants to generate detectable aerosols and only 29% of participants produced aerosols of <5 μm at visit 2 (at a median of 6 days after symptom onset), thus broadly in line with the heterogeneity of infectiousness as seen with other respiratory infections such as measles, influenza, and tuberculosis[18]. The variability outlined could not be explained by clinical or demographic variables (Table 1).

We then evaluated the relationship between probably highly infectious persons and patient and viral characteristics (Table 2). Nasopharyngeal Ct value was negatively associated with probably highly infectious persons (defined as being aerosol positive on visit 2 i.e., median of 6 days after symptom onset in contradistinction to moderately infectious persons [only aerosol positive on visit 1] and probably non-infectious [aerosol culture negative]). Three participants were initially tested with exhaled breath and no coughing, and subsequently completed the cough protocol. Two of the three were cough aerosol culture positive but all three were aerosol culture negative using the exhaled breath and no coughing protocol, which was abandoned in the early phase of the study (as performing both protocols was not pragmatic from a workload or cost point of view).

### Relationship between aerosol positivity and symptoms and viral genetic variants
Although respiratory symptoms (cough, sore throat, shortness of breath, chest pain) were associated with infectiousness (Fig. 2C), almost one-third of aerosol culture positive individuals had no respiratory symptoms or were asymptomatic, making this metric unreliable for guiding public health interventions. We further showed

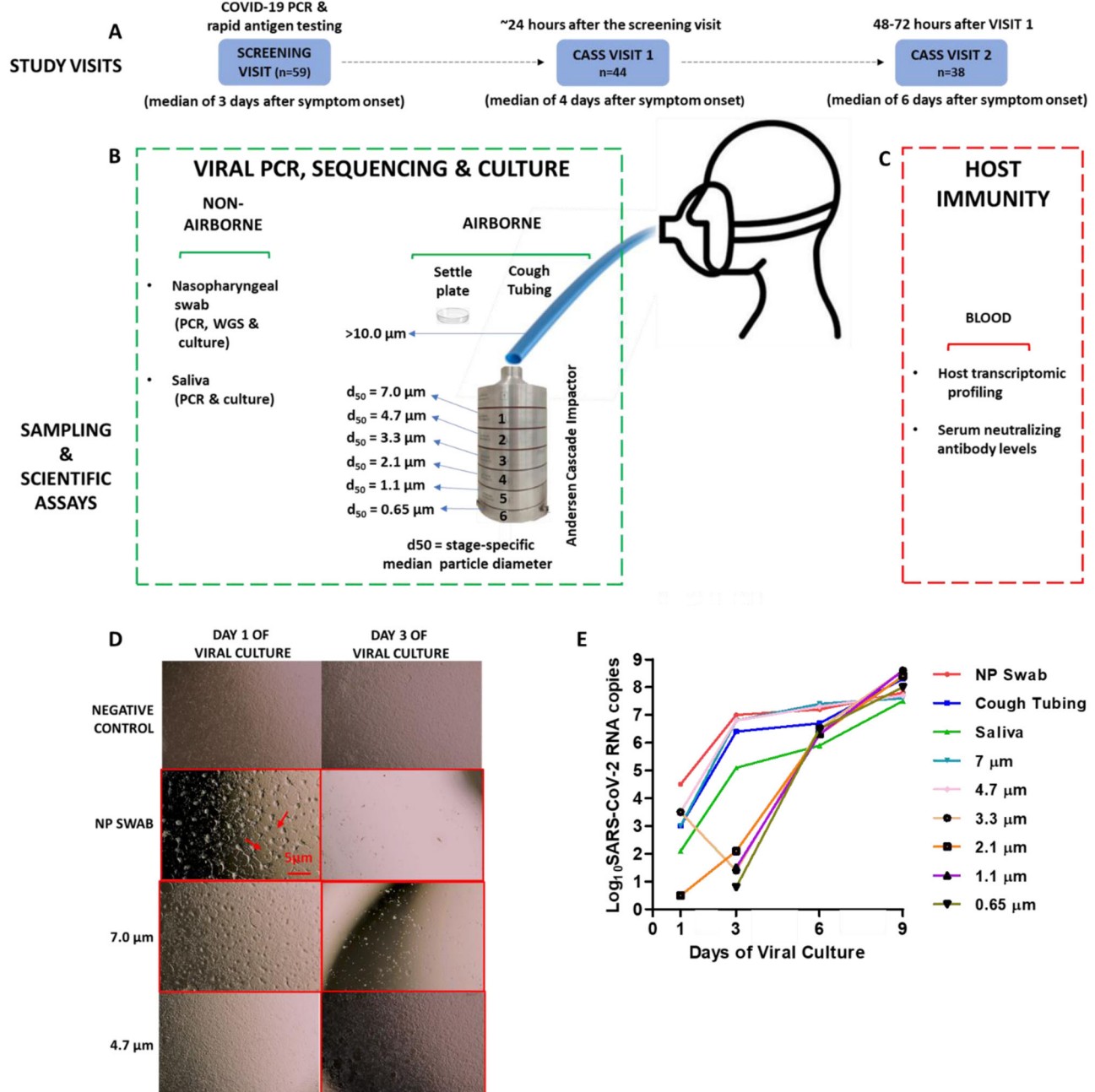

**Fig. 1 | Study overview and scientific experiments. A** Participants (*n* = 59) were screened at COVID-19 testing facilities and the next day underwent evaluation using the cough aerosol sampling system (CAS) if the PCR and/or rapid antigen test was positive and within 7 days of symptom onset (visit 1; *n* = 44). A follow-up visit was scheduled 48–72 h thereafter (visit 2; *n* = 38). **B** CAS was undertaken using a six-stage Anderson cascade impactor connected to cough tubing leading to a mouthpiece. A settle plate was included for collection of airborne SARS-CoV-2 in the cough cubicle (participant asked to count from 1–100 as loud as possible prior to cough sampling). Upper respiratory samples (nasopharyngeal swab, saliva) were also collected. **C** Venous blood to evaluate host transcriptomics and soluble bio-markers such as neutralizing antibodies were collected over a 48-hour period (visit 1 and 2). **D, E** All respiratory samples were assessed for SARS-CoV-2 by quantitative RT PCR and positive samples were further investigated for potential infectiousness by viral culture. Culture positivity was ascertained using a combination of cyto-pathic effect visualised using a light microscope (**D**) in tandem with confirmation of longitudinally increasing viral load determined by PCR (**E**). Representative micro-graphs are shown for baseline respiratory samples collected from participant CAS013, displaying cells at 24 h and 72 h post-infection. Samples showing the cytopathic effect (denoted by red arrow) have been highlighted by the bordered micrographs (clearing of the cellular monolayer). The viral supernatant was collected at 1, 3, 6 and 9-days post-infection and tested by qRT PCR. Viral culture positivity was defined as an increase, of at least 100-fold, in viral copy number over the 9-day duration of culture. Viral culture experiments were duplicated for 10% of all samples to assess reproducibility of the technique.

that the likelihood of infectiousness was highest within the first 8 days of symptom onset (Fig. 2D) in keeping with human lung challenge and other studies that conducted viral culture of samples from the upper respiratory tract[19]. Supplementary Fig. 1A and 1B depict the complete size fragmentation of sampled aerosols across reported respiratory symptoms and timing symptom onset (≤8 or >8 days) respectively. Supplementary Fig. 1C portrays the temporal pattern of aerosol culture status across categorization of symptoms. We could not definitively ascertain whether any specific viral variant (Beta, Delta versus Omicron) was associated with greater infectiousness (Fig. 3A), and found

no evidence that viral genomic variants including mutations in the envelope-encoding proteins (potentially portending ability to better withstand desiccation and UV light) could explain infectiousness (Table S1). This is likely a consequence of the limited sample numbers in each sub-group.

### Relationship between aerosol positivity and host immunity (neutralising antibodies and blood mRNA sequencing)

Next, we studied host immunity and its relationship to infectiousness. We found that the ability to aerosolise culturable virus was significantly associated with low variant-specific serum neutralizing antibody levels (Fig. 3B, C; Supplementary Figs. 1D, F).

The blood mRNA sequencing-related differential expression (DE) analysis (Table S5) identified a total of 11 upregulated and 9 downregulated genes when comparing CAS-positive to CAS-negative individuals (FDR < 0.05). By contrast, at an uncorrected $p$-value cut-off of <0.05, the same comparison identified 1735 upregulated and 1278 downregulated genes. The gene set enrichment analysis (GSEA) of the related gene ontology biological process pathways, identified significant enrichments ($p$-value < 0.05) in activated and suppressed pathways when comparing CAS-positive to CAS-negative individuals. Pathways significantly activated in CAS-positive individuals include responses to biotic stimuli including virus and bacteria, innate and inflammatory immune responses as well as production and responses to cytokines and type I interferons. Pathways suppressed in CAS positive individuals, included those related to complement and B cell activation, phagocytosis, and immunoglobulin mediated immune responses, and those related to ion transport, development, and neuronal sensory perception (Fig. 4A, B). Figure 4C displays an enrichment map illustrating the 30 most enriched and suppressed gene ontology biological process pathways when comparing CAS-positive to CAS-negative individuals. Supplementary Figs. 2A & 2B (tree plots) and 3A & 3B (ridgeplots) provide more detail on upregulated and downregulated genes when comparing aerosol culture positive to aerosol culture negative participants.

### Diagnostic predictors of infectiousness (nasopharyngeal swab Ct value and blood-based biosignature)

A number of biological patterns predicted infectiousness. Firstly, we found that a Ct value of <16.3 (using the CDC 2019-Novel Coronavirus (2019-nCoV) RealTime Reverse Transcriptase (RT)-PCR Diagnostic Kit) ruled in ~40% of aerosol culture-positive participants (i.e., modest sensitivity but very high specificity and thus high confidence that they were infectious; Fig. 3D, E, Table S2). The exclusion of participants above a Ct value of greater than 27 (negative predictive value) was less helpful in itself as it ruled out ~20% of the tested population as being highly likely to be non-infectious (i.e., high confidence that they were non-infectious). However, a combination of both could be useful to determine public health strategy in almost two thirds of persons (high confidence infectious and non-infectious persons). When time from symptom onset was combined with Ct value, rule-in value only marginally increased to 42.5%. A ROC (Supplementary Fig. 1E) for nasopharyngeal Ct value as a proxy for aerosol culture positivity was generated from the data included in Table S2. Of note, a strong association was observed between the capacity to produce culture positive aerosols and having either a culture positive nasopharyngeal swab, culture positive saliva or culture positive cough tubing sample (Table S3). Accuracy of nasopharyngeal swab Ct as a proxy for the identification of selected potentially highly infectious cases is summarised in Table S4 (and the ROC from the data plotted in Supplementary Fig. 4).

Predictive modelling using the transcriptional data identified a number of models that achieved high specificity and sensitivity using 2 and 3 gene blood biomarker combinations. Models that achieved a sensitivity and specificity >90% included the 3 biomarker

**Table 1 | Demographic and clinical characteristics of aerosol culture positive (10 μm and 5 μm) versus aerosol culture negative ambulatory persons**

| | All participants | Culture negative aerosol | Culture positive aerosol (<10 μm) | Culture positive aerosol (<5 μm) |
|---|---|---|---|---|
| $n$ (total visits = 82) | $N = 44$ | $N = 36$[a] | $N = 46$[a] | $N = 33$[a] |
| Sex (%) | | | | |
| Male | 21 (48) | 19 (53) | 20 (43) | 14 (42) |
| Female | 23 (52) | 17 (47) | 26 (57) | 19 (58) |
| Average age/ years (range) | 38 (20–71) | 37 (20–59) | 38 (21–71) | 39 (21–71) |
| BMI (%) | | | | |
| Normal | 22 (50) | 16 (44) | 24 (52) | 16 (48) |
| Overweight | 15 (34) | 13 (36) | 16 (35) | 13 (39) |
| Obese | 7 (16) | 7 (20) | 6 (13) | 4 (12) |
| Risk Factor (%) | | | | |
| None | 33 (75) | 30 (83) | 33 (72) | 22 (67) |
| Hypertension | 5 (11) | 3 (8) | 5 (11) | 5 (15) |
| Asthma | 3 (7) | 0 (0) | 6 (13) | 5 (15) |
| Diabetes | 1 (2) | 0 (0) | 1 (2) | 1 (3) |
| HIV | 1 (2) | 0 (0) | 1 (2) | 1 (3) |
| Anemia | 1 (2) | 2 (6) | 0 (0) | 0 (0) |
| Pregnancy | 1 (2) | 1 (3) | 1 (2) | 0 (0) |
| Current Smoker (%) | 10 (22) | 9 (25) | 9 (20) | 5 (15) |
| Vaccinated (%) | 13 (30) | 12 (33) | 11 (24) | 4 (12) |
| Symptomatic | | | | |
| Any (%) | 40 (91) | 28 (78) | 44 (96) | 32 (97) |
| Median Duration/ days (IQR) | 4 (3–6) | 6 (4–9) | 5 (3–6) | 5 (3–6) |
| Respiratory (%) | 38 (86) | 23 (64) | 41 (89) | 29 (88) |
| Median Duration/ days (IQR) | 3 (3–6) | 6 (3–8) | 5 (3–6) | 5 (3–6) |
| Non-respiratory (%) | 33 (75) | 24 (67) | 35 (76) | 25 (76) |
| Median Duration/ days (IQR) | 4 (3–6) | 5 (3–8) | 5 (3–6) | 5 (3–6) |
| Cough metrics | | | | |
| Median number of coughs (IQR)* | 182 (140–231) | 186 (154–231) | 177 (124–234) | 186 (119–244) |
| Median peak flow (IQR) | 318 (250–408) | 347 (262–443) | 310 (216–395) | 300 (208–408) |
| Median peak cough flow (IQR) | 380 (263–487) | 398 (304–500) | 360 (239–487) | 374 (263–501) |

[a]Visits 1 and 2 were aggregated.

*Gentle cough over a 10-minute period (coughing occurred for a period of 1 min with alternating periods of 1 min of rest i.e. total cough duration was 5 min).

No significant differences between tabulated variables were observed between the three groups. All the participants had an oxygen saturation over >95% at sea level.

combinations of MIR4323, TBL1XR1, PPP3CB and MIR4323, TBL1XR1, TRPM2 as did the 2 gene combination of TBL1XR1 and PPP3CB (Fig. 4D). A few other 2 biomarker combinations and single gene biomarkers achieved sensitivities >90% and specificities >80%.

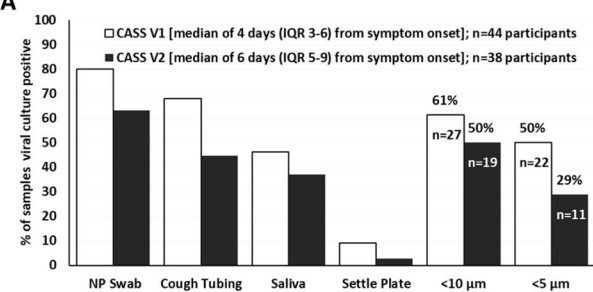

SARS-CoV-2 can be cultured from droplets & aerosols

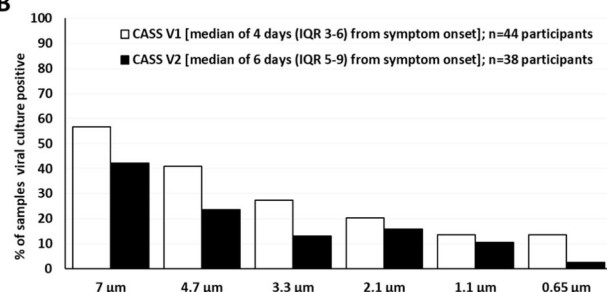

Size distribution of culture positive aerosols

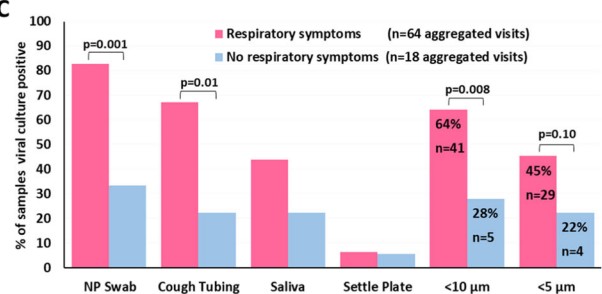

Association between respiratory symptoms & culture positivity

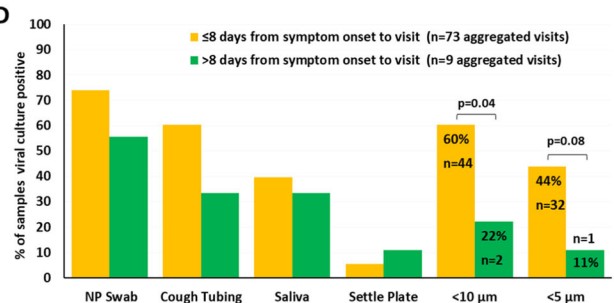

Association between symptom duration & culture positivity

**Fig. 2 | Determinants of aerosol culture positivity. A** Proportion of participants whose samples were culture-positive on non-aerosol (NP swab, cough tubing, saliva and settle plate) and aerosol (<10 μm and <5 μm) at visit 1 (☐) and visit 2 (■). **B** Proportion of size-fractionated aerosol samples that were culture-positive at visit 1 and visit 2. **C** Proportion of participants with (■) or without (■) respiratory symptoms producing culture-positive samples (NP swab, cough tubing, saliva and settle plate) and aerosol (visits aggregated, i.e visit 1 (64) + visit 2 (18) = 82 visits in total). **D** Proportion of participants producing culture-positive samples and aerosol based on duration from symptom onset to sampling ≤8 days (■) or >8 days (■) (visits aggregated, i.e., visit 1 (73) + visit 2 (9) = 82 visits in total). Viral culture positivity was defined as an increase, of at least 100-fold, in viral copy number over the 9-day duration of culture. The Fisher's exact test was used for comparisons between groups in (**C**) and (**D**).

## Discussion

To our knowledge, this is the most comprehensive report outlining the ability of COVID-19 patients to generate size-discriminated culturable virus in aerosol particles <10 μm, and to relate aerosol culturability to clinical, viral and host determinants. The frequency of aerosol virus culturability was high at ~60%. This is in contrast to published studies using size fractionated air sampling methodologies that, although few in number, have been mostly unsuccessful at culturing virus from aerosols generated from patients with acute SARS-CoV-2 infection[9,20–22] (with two studies demonstrating culturable virus in a total of five COVID-19 cases)[16,17] This could potentially be explained by several factors including viral dehydration and destruction during collection and laboratory transport, viral destruction due to impact forces related to the method of collection (e.g. high velocity impingement systems), and viral retention in sampling equipment and tubing[9]. To circumvent these hurdles, we used a gentle/low vacuum-based Andersen cascade impactor system (allowing graded multi-size discrimination) that we previously used to culture *M. tuberculosis* from size fractionated human aerosols[5] and (after various trial and error optimisation steps) we lightly sprayed agarose coated plates contained within the impactor with a thin film of viral culture medium, in tandem with rapid cold chain transportation to the laboratory to prevent viral dehydration.

There was considerable heterogeneity in the ability of participants to generate viable virus-laden aerosol and only ~30% of participants were able to produce aerosols of <5 μm at visit 2. Furthermore, about a third of patients were probably non-infectious, 50% probably highly infectious, and ~20% moderately infectious by our definitions. These data may be in line with the 'super-spreader' hypothesis, which assumes the presence considerable heterogeneity in infectious

probability with a limited proportion of persons being highly infectious driving a high proportion of transmission events[23,24]. This has been demonstrated with other respiratory viral and non-viral illnesses including measles, influenza, SARS-CoV-1, and tuberculosis[18]. There is no consensus on what constitutes infectiousness, but we reasoned that individuals who produced culturable aerosol for prolonged periods, in comparison to those who produced for limited periods, were more infectious. Notably, we did not unequivocally demonstrate transmission, but this is confounded by asymptomatic persons spreading COVID-19. There is no precise definition of what constitutes a 'super-spreader' in terms of aerosol culture positivity the term is not specific for infectiousness or transmission or both. Unfortunately, we were unable to use cell culture infectious viral burden as a metric for infectiousness because of sampling error (samples were collected on the impactor plates containing variable amounts of viral material [and not in liquid format – the latter allowing precise quantification per unit volume]; furthermore, unlike in bacterial and mycobacterial infections there are no observable colonies and thus no colony counts could be ascertained)[5]. The two other studies which have managed to culture SARS-CoV-2 from bioaerosols were similarly unable to quantify infectious viral burden and potentially the degree of infectiousness[16,17]. It is unclear where the infectious aerosol is locationally and anatomically being generated within the respiratory tract. Patients who were NP swab culture-positive varied considerably in their ability to produce infectious aerosols (Table 2). We speculate that such persons may have had more extensive infection of the upper respiratory tract and large airways, rather than the small airways and alveoli (as none required supplemental oxygen making alveolitis/ pneumonia unlikely).

Almost a third of individuals were aerosol culture-positive for particles <10 μm despite having no respiratory symptoms (almost a

**Table 2 | Heterogeneity in aerosol culture positivity (probable infectiousness) in those for whom aerosol culture data was present for both visits (n = 38)**

| | Probably highly infectious | Probably non-infectious (culture -ve) | OR (95% CI)* | p-value | Probably moderately infectious | OR (95% CI) vs probably highly infectious◇ | p-value | OR (95% CI) vs probably non-infectious‡ | p-value |
|---|---|---|---|---|---|---|---|---|---|
| n (% of total) | 19 (50) | 12 (32) | | | 7 (18) | | | | |
| Median NPS Ct (IQR)† | 16.9 (14.5–21.8) | 24.1 (19.0–34.9) | 1.2 (1.0–1.4) | **0.01** | 19.1 (11.3–20.8) | 1.0 (0.7–1.2) | 0.93 | 1.2 (1.0–1.5) | 0.12 |
| Culture positive NPS (%)† | 18 (95) | 7 (58) | 12.9 (1.3–131) | **0.02** | 5 (71) | 7.2 (0.6–96.7) | 0.17 | 1.8 (0.2–13.2) | 0.66 |
| Median cough tubing Ct (IQR)† | 20.5 (17.4–24.9) | 26.8 (24.0–37.9) | 1.2 (1–1.3) | **0.003** | 23.3 (22.9–30.1) | 1.2 (1.0–1.5) | 0.06 | 1.0 (0.7–1.2) | 0.35 |
| Culture positive cough tubing (%)† | 16 (84) | 4 (33) | 10.7 (1.9–59.4) | **0.007** | 5 (71) | 2.1 (0.3–16.6) | 0.59 | 5.0 (0.7–38.2) | 0.17 |
| Respiratory symptoms at visit (%)† | 15 (79) | 7 (58) | 2.7 (0.5–13.2) | 0.25 | 5 (71) | 1.5 (0.2–10.8) | 0.65 | 1.8 (0.2–13.2) | 0.66 |
| Median duration respiratory symptom/days (IQR)† | 3 (3–4.3) | 6 (2.5–6) | 1.4 (0.9–2.2) | 0.27 | 3 (2.5–5) | 1.1 (0.5–2.8) | 1 | 1.0 (0.5–2.2) | 0.38 |
| Median number of coughs (IQR)† | 169 (121–235) | 186 (158–229) | 1.0 (1.0–1.0) | 0.48 | 208 (163–235) | 1.0 (1.0–1.0) | 0.33 | 1.0 (0.9–1.0) | 0.64 |
| Sex (%) | | | 0.4 (0.1–1.8) | 0.29 | | 0.4 (0.1–2.6) | 0.41 | 1.0 (0.1–6.3) | 1.00 |
| Male | 7 (37) | 7 (58) | | | 4 (57) | | | | |
| Female | 12 (63) | 5 (42) | | | 3 (43) | | | | |
| Median Age/years (IQR) | 36 (29–42) | 36 (27–50) | | 0.98 | 33 (27–40) | | 0.58 | | 0.67 |
| BMI (%) | | | | | | | | | |
| Normal | 9 (50) | 6 (50) | | | 3 (43) | | | | |
| Overweight | 7 (39) | 3 (25) | 0.6 (0.1–3.5) | 0.69 | 4 (57) | 1.7 (0.3–10.3) | 0.67 | 0.4 (0.05–2.9) | 0.61 |
| Obese | 3 (11) | 3 (25) | 1.5 (0.2–10.1) | 1.00 | 0 (0) | 1.0 (0.07–13.7) | 1.00 | 1.5 (0.11–21.3) | 1.00 |
| Risk Factor (%) | | | 1.8 (0.3–11.1) | 0.68 | | 2.1 (0.2–22.5) | 0.65 | 0.8 (0.06–11.3) | 1.00 |
| Yes | 5 (26) | 2 (17) | | | 1 (14) | | | | |
| No | 14 (74) | 10 (83) | | | 6 (86) | | | | |
| Vaccinated (%) | 4 (27) | 4 (33) | | | 2 (29) | | | | |
| Viral (%) | | | | | | | | | |
| Delta | 13 (68) | 5 (42) | | | 6 (86) | | | | |
| Beta | 5 (26) | 4 (33) | 2.1 (0.4–11.1) | 0.42 | 0 (0) | 0.4 (0.04–4.6) | 0.64 | 4.8 (0.4–58.1) | 0.31 |
| Omicron | 1 (6) | 3 (25) | 7.8 (0.6–93.9) | 0.12 | 1 (14) | 2.2 (0.1–40.8) | 1.00 | 3.6 (0.3–46.4) | 0.57 |

Non-infectious (NI) = CAS negative at both visits (median of D3 and 6 after symptom onset; n = 12); probably highly infectious = CASS positive at visit 2 (n = 19); probably moderately infectious = CAS positive at one visit but not visit 2 (n = 7). Note: n = 6 only attended visit 1. For the purposes of calculating median Ct value only visit 1 was used as this best reflected Ct at or close to diagnosis. The Fisher's exact test was used for comparisons involving categorical outcomes, Mann-Whitney test was used for non-parametrically distributed continuous data.
*Probably highly infectious as reference group.
◇Probably highly infectious as reference group.
‡Probably moderately infectious as reference group.

quarter for particles <5 µm). Thus, consistent with other studies asymptomatic persons may be nasopharyngeal swab positive and infectious[25,26]. However, here we show for the first time that asymptomatic persons may also produce infectious aerosol <10 µm which is potentially suspensible for several hours and may be deeply inoculated by inhalation into the small airways and alveoli of the lung. Irrespective of particle size (0.65 µm to 7 µm) there was a clear relationship between increasing time from symptom onset and decreasing aerosol culturability, though ~10% of individuals continued to emit infectious aerosol more than 8 days after symptom onset, again demonstrating considerable heterogeneity (discussed above).

We could not definitively ascertain whether any specific variant (Beta, Delta versus Omicron) was associated with greater infectiousness. This is likely a consequence of the limited sample numbers in each sub-group. Reported epidemiological data indicate that vaccination is associated with reduced transmission of SARS-CoV-2 (albeit less effectively with the Omicron variant);[27] again, our sample size was inadequate to definitively address the question of whether, and to

what extent, vaccination limits aerosolization capability. Although a higher proportion of culture negatives were vaccinated, we were not powered to detect small differences in effect size. Similarly, our study was not powered to identify genetic variants more likely to be associated with infectiousness.

How might these data impact clinical practice and public health policy? First, consistent with prior work that evaluated NP culture positivity (but not aerosol positivity)[26], we found that nasopharyngeal Ct values correlated with the likelihood of infectiousness, and a Ct value < 17 ruled in ~40% of aerosol culture positive persons with a very high positive predictive value (almost 100%). A Ct value > 27 effectively ruled out ~25% of the tested population. Thus, in almost two-thirds of persons, nasopharyngeal Ct values (a readily available metric from PCR tests) could direct further clinical and public health interventions. Second, we found that a simple 3-transcript blood-based biosignature was highly predictive of emitting culturable virus (PPV > 90%). This readout could easily be ascertained using a multiplex PCR assay that could be automated within a point of care cartridge-based format (e.g.,

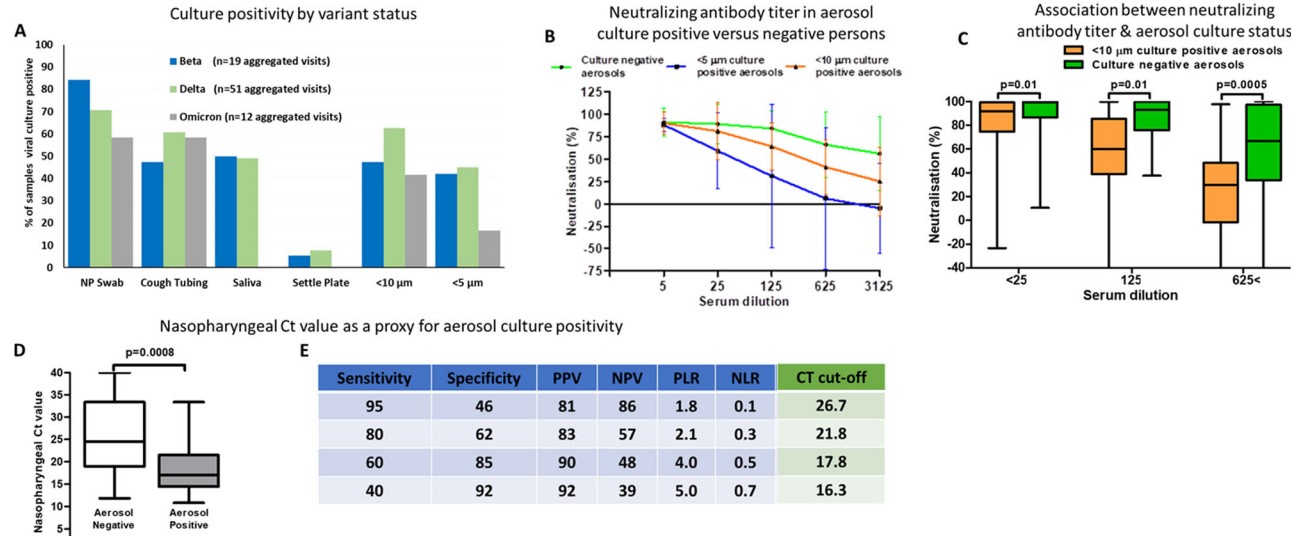

**Fig. 3 | Virologic and immunological determinants of aerosol culture positivity.**
**A** Proportion of samples that were culture-positive in non-aerosol and aerosol (<10 μm and <5 μm) for the Beta (■), Delta (■) or Omicron (■) variants (visits aggregated, i.e., visit 1 + visit 2 = 82 visits in total). **B** Neutralization of SARS-CoV-2 pseudovirus by patients' sera (*n* = 18 for negative, *n* = 7 for <5 μm; *n* = 7 for <10 μm). Data are presented as median values +/− SEM. **C** Neutralization activity in aerosol culture positive and negative persons (mean of each group with standard deviation as error bars; *n* = 18 for negative, *n* = 7 for <5 μm; *n* = 7 for <10 μm). **D** Nasopharyngeal Ct values in the aerosol culture negative (*n* = 13) and positive participants (*n* = 31); NP Ct values were used from visit 1 (except in 4 participants who were culture +ve on visit 2 but not visit 1). **E** Performance outcomes of nasopharyngeal Ct as a predictor for culture positive aerosol (sensitivity, specificity etc. expressed for various Ct cut-points incorporating rule-in, rule-out and Youden's index readouts). The neutralization capacity (**C**) and nasopharyngeal Ct (**D**) are depicted by box-and-whisker plots indicating the median (middle line), 25th (bottom line) and 75th percentiles (top line), and the range (whiskers) of the measured parameters. Mann-Whitney test was used for comparisons across groups in (**C**) and (**D**).

Xpert platform; Cepheid) to rapidly determine the likelihood of infectiousness. This might guide isolation periods and the need thereof, in certain professions with serious implications for transmission (e.g., health care and nursing home workers, airline workers, teachers, policemen etc) or where there might be concern about exposing elderly or immunocompromised persons in the home setting. To our knowledge this is the first proof-of-concept that a host transcriptomic signature can predict SARS-CoV-2 infectiousness. We are unsure whether this might apply to other respiratory viruses too (subject of an ongoing study). The 3 genes highly predictive of infectiousness were those encoding transcripts involved in RNA regulation (for example of CXCL10 a key Th1 chemokine implicated in the COVID-19 hyperinflammatory syndrome or cytokine storm[28]), transcription factor activation, and immune function suggesting that host pathogen relationships are important in the heterogeneity of infectiousness. Thus, targeted interventions could be applied to individuals identified in this way (using NP Ct value and/or multiplex PCR signature) to target or implement oral drug therapy such as nirmatrelvir/ritonavir (Paxlovid), contact precautions, quarantine, improved ventilation, shielding of immune-vulnerable persons, and targeted isolation in workplace settings (e.g., workers in healthcare facilities and retirement homes, etc.). These findings are still relevant and important as although serious illness with newer variants such as Omicron has declined, mortality remains significant in vulnerable populations and in some countries and settings such as the UK and China.

The blood transcriptomic analysis revealed that several sets of pathways related to type 1 interferon, viral defence mechanisms, cytokine production/signalling, and adaptive immunity were predictably upregulated in those with culturable aerosol. However, specific innate immune pathways (those associated with complement[29] and its activation) and pathways associated with suppression of humoral immunity (immunoglobulin production and B-cell immunity/receptor signalling) were also associated with aerosol culture positivity. This supported our findings of low neutralising blood antibody

levels being associated with increased aerosol culturability. Interestingly, we found that pathways associated with nervous system processes, synaptic, and transsynaptic signalling were also relatively suppressed in those that were probably infectious compared to those that were probably non-infectious. This raises the possibility that neuroimmune modulation may impact the ability to produce aerosolised virus. It has recently been demonstrated that viral antigen can diffusely seed the central nervous system[30]. Neuro-immuno-modulatory mechanisms, intriguingly, can be effected through vagal efferents innervating the respiratory tract[31] and this was associated with modulation of airway inflammation and immunity[32]. Our data support the contention that host-pathogen interactions, including mucosal immunity, are critically important in determining host infectiousness. Thus, the higher demonstrated transmissibility of the Omicron variant in epidemiological studies may be related not to the ability to aerosolise virus, but due other factors including differential host pathogen interactions including heterogeneity in receptor binding, NK cell responses, innate immunity, type 1 interferon responses, and adaptive immunity[33], the infecting dose, etc.

There are several limitations of our study findings. Firstly, the study was confined to a South African population and of limited sample size, though this remains the largest study of this kind and the procedures involved were substantially labour and resource intensive. Second, ability to aerosolise virus is only one dimension of infectiousness and we may have underestimated the extent of infectiousness due to viral destruction during transport and inability to culture the virus due to technical factors (such as endogenous bacterial/fungal contamination during viral culture). There was some sampling variability (a few patients were culturable on visit 2 but not visit 1). We may have also overestimated infectiousness as the particles were directly and rapidly collected on size-fractionated plates rather than being exposed to the environment for an extended time period; however, we also cultured virus from settle plates. Third, the transcriptomic biosignature predicting probably infectiousness was not validated in an

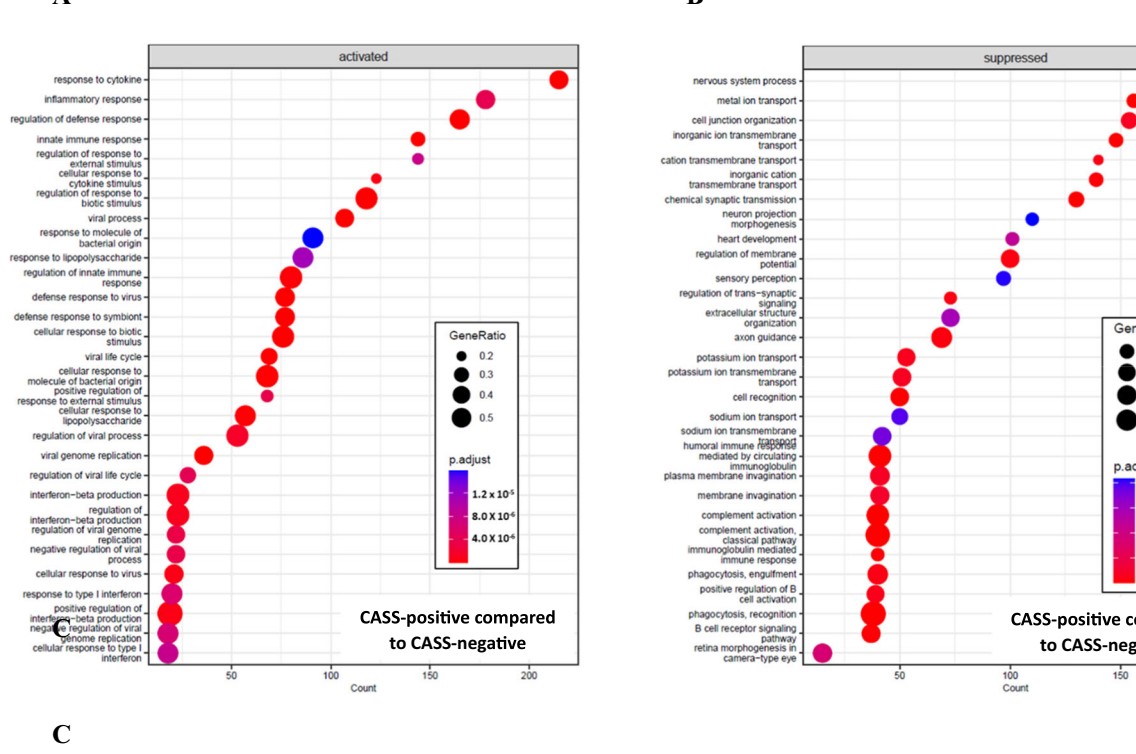

**D**

| | MIR4323, TBL1XR1, PPP3CB | MIR4323, TBL1XR1, TRPM2 | MIR4323, TBL1XR1 | TBL1XR1, PPP3CB | MIR4323, PPP3CB | TBL1XR1, TRPM2 |
|---|---|---|---|---|---|---|
| **Probability cut-off** | 0.5 | 0.5 | 0.5 | 0.5 | 0.5 | 0.5 |
| **Sensitivity** | 0.95 (0.76-1.00) | 0.9 (0.70-0.99) | 0.9 (0.70-0.99) | 0.95 (0.76-1.00) | 0.95 (0.76-1.00) | 0.9 (0.70-0.99) |
| **Specificity** | 0.91 (0.59-1.00) | 0.91 (0.59-1.00) | 0.82 (0.48-0.98) | 0.91 (0.59-1.00) | 0.82 (0.48-0.98) | 0.82 (0.48-0.98) |
| **PPV** | 0.95 (0.76-1.00) | 0.95 (0.75-1.00) | 0.9 (0.70-0.99) | 0.95 (0.76-1.00) | 0.91 (0.71-0.99) | 0.9 (0.70-0.99) |
| **NPV** | 0.91 (0.59-1.00) | 0.83 (0.52-0.98) | 0.82 (0.48-0.98) | 0.91 (0.59-1.00) | 0.9 (0.55-1.00) | 0.82 (0.48-0.98) |
| **AUC** | 0.978 | 0.974 | 0.965 | 0.978 | 0.961 | 0.913 |

independent cohort and thus we may have 'over-fitted' the data. However, we performed a bootstrapping analysis that incorporated a training and test set principle, and thus we provided preliminary proof-of-concept that a host biosignature when coupled to positive diagnostic testing could serve a proxy of probable infectiousness. Fourth, our data were generated from patients who were asked to gently

cough into the CAS system and we were unable to report comprehensively on tidal breathing, talking and forced respiratory manoeuvres. However, we did undertake tidal breathing-based analysis in 3 participants (we were unable to culture virus in these 3 participants on tidal breathing but 2 of the 3 were cough aerosol positive) but this approach was abandoned because of resource constraints and is

**Fig. 4 | Transcriptomic profile predicting potential infectiousness.** Dot plot of gene set enrichment analyses illustrating the 30 most enriched (**A**) and suppressed (**B**) gene ontology biological pathways, when comparing CAS-positive to CAS-negative individuals. The *x*-axis represents the number of enriched genes detected in each pathway. The dot size represents the gene ratio, which is the proportion of all genes in the pathway that were determined to be enriched or suppressed. The dot color represents the adjusted *p*-value as indicated in the legend. In summary, activated pathways in CAS- positive individuals included responses to biotic stimulus including virus and bacteria, innate and inflammatory immune responses as well as production and responses to cytokines and type I interferon. Pathways that are suppressed in CAS positive individuals included complement and B cell activation, phagocytosis and immunoglobulin mediated immune responses as well as pathways related to ion transport, development and neuronal sensory perception. **C** Enrichment map illustrating the 30 most enriched and suppressed gene ontology biological process pathways when comparing CAS-positive to CAS-negative individuals. Enriched terms are clustered into networks with edges linking gene sets with overlapping terms to identify functional modules. For activated pathways, three overlapping clustered functional modules can be identified including (i)

terms related to inflammatory/innate immunity and responses to cytokines (red circle), (ii) viral defence responses including production and responses to type I interferons (blue circle) as well as (iii) pathways involved in bacterial defence responses (green circle). For repressed pathways three distinct functional modules are present including (i) pathways involved in phagocytosis, humoral immunoglobulin immunity, complement and B cell activation and receptor signaling (red circle), (ii) ion transport and neuronal signaling (blue circle) and (iii) distinct pathways involved in development/morphogenesis (green circle). The edges (grey line) link pathways with overlapping terms with shorter lengths indicating greater similarity. The dot size represents the number of enriched genes in the pathways while color represents adjusted p-values as indicated in the legend. **D** Selection of genes best predicting infectiousness as measured by capacity of patients to emit culture positive aerosols. PPV Positive predicting value, NPV Negative predictive value, AUC Area under curve, MIR4323 microRNA 4323, TBL1XR1 transducing (beta)-like 1X-linked WD40 repeat-containing gene, PPP3CB protein phosphatase 3 catalytic subunit beta, TRPM2 transient receptor potential cation channel, subfamily M, member 2. An *F*-test was used to compare CAS-positive to CAS-negative participants.

planned for a follow-up study. Fifth, we employed a cough manoeuvre that may be viewed as intensive (coughing for 50% of 10 min). However, studies have demonstrated that a high proportion of symptomatic patients with COVID-19 have cough as a symptom[34] and when the total daily duration of cough is factored in, the duration we used may be biologically meaningful. Finally, whilst we are able to potentially infer infectiousness of COVID-19 positive persons, we cannot make any conclusions about transmission as close contacts of study participants were not evaluated (thus we did not look at epidemiological evidence of onward transmission mainly because this is heavily confounded by transmission by asymptomatic persons). Although we had accessed a proportion of the contacts, attack rates in close contacts are known to be only in the region of 20 to 30% (based on detectable virus in nasopharyngeal swabs), and any viral-specific memory responses in such persons could have been from previous unrelated episodes of exposure. Thus, inferring transmission to a particular individual (a contact), at a particular point in time, is challenging if not impossible. Interestingly, there was a clear gradient of decreasing nasopharyngeal swab culture positivity (and PCR Ct value, and culture positivity in the cough tubing) across the probably infectious, moderately infectious, and non-infectious groups, suggesting that our chosen classification was likely biologically meaningful.

In summary, our findings indicated that SARS-CoV-2 is culturable from human-generated aerosol <10 μm in almost 60% of affected of the patients evaluated, although some individuals had the capability to aerosolise culturable virus in smaller particles for extended time periods. The ability to aerosolise culturable virus is heterogenous and inversely related to time from symptom onset and modulation of specific components of host immunity. Nasopharyngeal Ct thresholds and a preliminary unvalidated blood-based biosignature was highly predictive of infectious aerosol generation. These data support the need for prevention of airborne transmission risk using better ventilation in public transports, and indoor environments, especially hospitals, workplaces and schools, and the use of other airborne infection controls in health care facilities caring for COVID-19 patients.

## Methods

### Study design, eligibility criteria and regulatory approvals

A diagnostic study, approved by the Human Research Ethics Committee at the University of Cape Town (HREC REF: 387/2020), was used to identify and recruit participants into this study. Informed consent was obtained from all participants included in this study. We used a cross-sectional study design with prospective follow-up. Eligibility criteria included being ≥18 years, testing positive for COVID-19 by PCR or rapid antigen, ambulatory, being within 7 days of symptom onset,

having a blood oxygen saturation ($O_2$ SAT) ≤95 and willing to undergo CAS and provide nasopharyngeal and saliva specimens. All laboratory aspects of this study were approved by the University of Cape Town Institutional Biosafety Committee prior to study initiation. SARS-CoV-2 culture was performed in a BSL-3 laboratory by vaccinated personnel utilizing powered air purifying respirators.

### Patient clinical information, sample collection and CAS procedure

From January 2021 to May 2022, we recruited 44 ambulatory patients with PCR-confirmed COVID-19 who underwent CAS (designated visit 1; Fig. 1A). A second CAS sampling was undertaken ~48 to 72 h after visit 1 (designated visit 2; 38 participants completed this visit; Fig. 1A). Staff used N95 masks and wore comprehensive personal protective equipment. Consenting patients underwent a nurse-administered clinical and sociodemographic interview. Sex of every participant was captured as part of this interview and was a self-reported status. Peak expiratory flow (PEF), forced expiratory volume (FEV) and cough peak flow (CPF) were measured using an Asma-1 Electronic Respiratory Monitor (Vitalograph, United Kingdom) and a Respi-Aide Peak Flow Meter (GaleMed, China). The average of three consecutive blows were used. All procedures were done at one visit in the same order.

A nasopharyngeal swab in viral transport medium (VTM), saliva samples and blood for host immunity (Fig. 1C) interrogation were collected prior to aerosol sampling. Viral transport medium (VTM) was prepared in the laboratory using Anderson's modified Hanks Balanced Salt Solution (8.0 g/l NaCl, 0.4 g/l KCl, 0.05 g/l Na2HPO4, 0.06 g/l KH2PO4, 1.0 g/l Glucose, 0.7 g/l NaHCO3, 0.2 g/l MgSO4.7H2O, 0.14 g/l CaCl2.2H2O) with 2% v/v heat-inactivated fetal calf serum, 100 μg/ml gentamicin, 100 I.U/ml penicillin, 100 μg/ml streptomycin and 2.5 μg/ml of Amphotericin B.

CAS was performed in a negative pressure well-ventilated sampling cubicle embedded within a larger negative pressure room. The CAS apparatus consisted of a facemask attached to tubing connected to a vacuum-based Anderson cascade impactor (see Fig. 1B) within a metal drum, with the flow rate monitored using a flow meter and kept at around 30 L/min[4]. Aerosol sampling was conducted using viral medium coated agar plates located at 6 stages (collects aerosol in a particle size-dependent manner) within the Anderson impactor, a settle plate at head level, placed 50 cm away from participant within the cubicle. Briefly, patients coughed as forcefully and as frequently as possible into the CAS for 5 min (5 sequences of 1 min cough followed by 1 min rest) via a 1 m silicone pipe that ran from the patient in a sputum induction booth into the CAS. Ambient temperature and humidity and the number of coughs were recorded.

## CAS and sample processing

The CAS was transported back to the laboratory under cold condition (maintained using ice packs) as soon as sampling was completed. Longest time between end of sampling and delivery of system was 45 min. The impactor was removed from its encasing drum in a biological safety cabinet within the BSL3 facility. Plates were sequentially removed, irrigated each with 1.2 ml of cell culture medium, with the plates carefully and gently swirled to ensure irrigated medium swept across the full area of the agar layer. Around 900–1000 µl of media was recovered and frozen down into 3 separate aliquots for downstream PCR and viral culture. Settle plate was treated in a similar way. Nasopharyngeal swab was vortexed for 5 s and VTM aliquoted and stored at −80 °C. 1.2 ml of cell culture media was added to the saliva specimen, mixed, aliquoted, and stored at −80 °C. PAXgene blood was left at room temperature overnight and frozen at −80 °C until RNA extraction. Serum was collected after centrifugation at $500 \times g$ for 10 min, aliquoted and stored at −80 °C until neutralization assay.

## Polymerase chain reaction of raw samples and viral culture supernatant

Nucleic acid amplification test using the Emergency use authorization assay (Catalog # 2019-nCoVEUA-01) developed by the USA Centers for Diseases Control and Prevention (CDC) was used to detect SARS-CoV-2 in respiratory samples and viral culture supernatant[35]. The panel was designed for specific detection of SARS-CoV-2 (two primer/probe sets targeting the nucleocapsid gene). An additional primer/probe set to detect the human RNase P gene (RP) in control samples and clinical specimens was also included in the panel. Results were classified as positive for SARS-CoV-2 when both the N1 and N2 targets of the nucleocapsid gene were detected by PCR with cycle threshold (Ct) values were <40.

## Viral culture

To establish the in vitro viral culture model, a SARS-CoV-2 viral stock (isolated from a COVID-19 patient during the Beta wave) was used to infect the human lung carcinoma cell line, H1299 (ATCC CRL-5803), in a BSL3 laboratory and infection was confirmed by light microscopy (as assessed by cytopathic effects of the virus on the cell line) and confocal microscopy (Fig. 1D). Serial dilutions of the viral stock were used to establish the limit of detection of the PCR assay at $1 \times 10^1$ copies/ml. Virus isolation culture was attempted from the nasopharyngeal swab, saliva, cough tubing, settle plates and CAS plates. The cell line was maintained in Roswell Parks Memorial medium (RPMI) containing 10% bovine serum, 100 IU penicillin/streptomycin, 2 Mm L-glutamine, 25 Mm HEPES, 1x non-essential amino acids and 0.1 mg/ml sodium pyruvate (ThermoFisher, South Africa). All sample aliquots destined for viral culture were initially filtered through a 0.22 µm filter prior to inoculation. The samples were diluted 2-folds using cell culture media during the filtration step. 250 µl of filtered samples were inoculated in respective wells and cultures were grown in a humidified 37 °C incubator with 5% $CO_2$ and cytopathic effect (CPE) (Fig. 1D) and viral replication (Fig. 1E) were monitored on days 1, 3, 6 and 9 by PCR. Viral culture positivity was defined as at least a 100-fold increase in viral load over time. Reproducibility of the viral culture assay was ascertained through the duplicate culture of a subset of samples (nasopharyngeal swab and CAS plates), in separate culture experiments. Replicability was defined as similar qualitative outcomes, that is, culture positive or negative in both runs.

## SARS-CoV-2 whole genome sequencing and genome assembly

Viral sequencing was performed through the Network for Genomics Surveillance in South Africa (NGS-SA), at the University of Cape Town Division of Medical Virology. RNA was extracted from harvested viral-culture supernatant (Vero E6 or H1299 cells) on an automated Maelstrom™ 4800 using the TANBead® Nucleic Acid Extraction Kit

(Taiwan Advanced Nanotech Inc, Taipei, Taiwan) or Chemagic™ 360 automated system (PerkinElmer, Inc, Waltham, MA) as per manufacturer's protocol. Whole genome amplification and library preparation were performed using the Illumina COVIDSeq Test kit and protocol 1000000128490 v02 (Illumina, Inc., San Diego, CA), and executed on the Hamilton Next Generation StarLet (Hamilton Company, Reno, NV). Whole genome amplification was achieved via multiplex polymerase chain reaction performed with the ARTIC V4.1 primers designed to generate 400-bp amplicons with an overlap of 70 bp that spans the 30 kb genome of SARS-CoV-2. Indexed paired-end libraries were normalized to 4 Nm concentration, pooled, and denatured with 0.2 N sodium acetate. A 4 Pm pooled library was spiked with 1% PhiX Control v.3 adaptor-ligated library (Illumina, Inc., San Diego, CA) and sequenced using the MiSeq® Reagent Kit v2 (500 cycle) and sequenced on the MiSeq instrument (Illumina, Inc., San Diego, CA).

The quality of sequencing reads was assessed using different tools including FastQC, Fastp, Fastv, Fastq_screen, and Fastx_toolkit. The resulting reads were analyzed on Exatype (https://exatype.com/) for referenced-based genome assembly and to identify minor and major variants. The assembled consensus sequences were analyzed using Nextclade Web (https://clades.nextstrain.org) for further quality control and clade assignment. The Stanford Coronavirus Resistance Database (CoV-RDB; https://covdb.stanford.edu), designed to house comprehensively curated published data on the neutralizing susceptibility of SARS-CoV-2 variants and spike mutations to monoclonal antibodies, convalescent plasma, and vaccinee plasma, was used in mutational analysis to identify relevant mutations in the sequenced viruses and allow for comparative assessment of their occurrence across different phenotypes of participants[36]. Accession codes for all sequencing data are provided in Table S6.

## Neutralization assay

The 293 T/ACE2.MF cells modified to overexpress human ACE2 were kindly provided by Dr Mike Farzan, Scripps Research. These cells were cultured in DMEM containing 10% heat-inactivated fetal calf serum and 3 µg/ml puromycin at 37 °C, 5% $CO_2$. Cell monolayers were disrupted at confluency by treatment with 0.25% trypsin in 1 Mm EDTA. SARS-CoV-2 pseudotyped lentiviruses were prepared by co-transfecting the HEK293T cell line with either the SARS-CoV-2 Beta spike (L18F, D80A, D215G, K417N, E484K, N501Y, D614G, A701V, 242-244del) or the Delta spike (T19R, R158G L452R, T478K, D614G, P681R, D950N, 156-157 del) plasmids in conjunction with a firefly luciferase encoding lentivirus backbone plasmid. For the neutralization assay, heat-inactivated plasma samples from participants were incubated with the SARS-CoV-2 pseudotyped virus for 1 h at 37 C, 5% $CO_2$. Choice of pseudotyped lentivirus system used (Beta or Delta spike) was based on autologous host-variant pairing, that is, participants infected with the Beta variant had their serum tested with the Beta spike, and likewise, participants infected with the Delta variant had their serum tested with the Delta spike. Subsequently, $1 \times 10^4$ HEK293T cells engineered to overexpress ACE-2 were added and incubated at 37 C, 5% $CO_2$ for 72 h upon which the luminescence of the luciferase gene was measured. CB6 was used as a positive control. Neutralization was measured as described by a reduction in luciferase gene expression after single-round infection of 293 T/ACE2.MF cells with spike-pseudotyped viruses. Titers were calculated as the reciprocal plasma dilution (ID50) causing 50% reduction of relative light units. All assays were run in duplicate.

## Blood host transcriptomics

Blood RNA sequencing was performed on 33 individuals (21 CAS-positive and 12 CAS-negative) using blood collected over a 48-hour period (visit 1 and 2; Fig. 1A). In brief, total RNA was extracted from PAXgene blood RNA tubes using the PAXgene blood RNA isolation kit (Qiagen, PreAnalytix; catalog: 762174). Sequencing libraries were

prepared using RNA and globin depletion and sequenced on the Illumina platform using 150 bp pair-end sequencing paired-end reads.

The FastQC program (version 0.11.9, https://www.bioinformatics.babraham.ac.uk/projects/fastqc/), was used to assess read quality and trimming was performed using the Trim Galore program (version 0.6.10, https://www.bioinformatics.babraham.ac.uk/projects/trim_galore/). The Spliced Transcripts Alignment to a Reference (STAR) software (version STAR_2.7.7a) (PMID: 23104886) was used to map reads to the Ensembl (PMID: 33137190) human genome primary assembly (version GRCh38.99) with the quantMode and GeneCounts option selected to generate raw genewise read counts for each sample.

The DE analysis was performed with the edgeR (version 3.38.4) (PMID: 19910308) Bioconductor package. Briefly, raw counts were filtered to remove genes with low expression, normalized, and negative binomial generalized linear models were fitted. The quasi-likelihood F-test was used to identify DE genes when comparing CAS-positive to CAS-negative individuals.

A gene set enrichment analysis (GSEA) for Gene Ontology (Biological Process) was performed on the differential expression results ranked by fold change using the gseGO function, from the clusterProfiler (ver: .4.4.4, PMID: 34557778) package in R. The simplify function was used to reduce redundancy in the enriched pathways.

**Predictive modeling.** Predictive modeling was performed on the transcriptional data using the random forest ranger algorithm in the tidymodels program (version, 1.0.0) in R. The 5 most important variables were identified from the 100 genes with the smallest p-values from the differential expression analysis. Predictive modelling was subsequently performed with 1000 bootstraps using various combinations of the top 5 most important variables and the best models were selected that used 2 to 3 gene biomarkers.

**Statistical analysis**

The Fisher's exact test was used for comparisons involving categorical outcomes, Mann-Whitney test was used for non-parametrically distributed continuous data, and logistic regression performed to generate odds ratios (Stata version 17 or GraphPad, Version 9.4.1). A p-value of <0.05 was considered significant for all statistical analyses, all tests were two-tailed and adjustment for multiple comparisons was limited to the transcriptomic analysis. Receiver operating characteristic (ROC) curves were generated for sensitivity/specificity analysis. The DE results were sorted/ranked by fold change and a gene set enrichment analysis (GSEA) for Gene Ontology (Biological Process) and KEGG pathways was performed using the gseGO and gseKEGG functions respectively, from the clusterProfiler (ver: .4.4.4 package in R. Pathways with an FDR < 0.05 were considered significant.

**Reporting summary**

Further information on research design is available in the Nature Portfolio Reporting Summary linked to this article.

# Data availability

All data generated and analyzed in this study are included in the paper and its Supplementary section. Individual participant data will be made available to researchers who provide a protocol that is approved by their respective human research ethics committee. All protocols will be reviewed and approved by the CAS COVID consortium trial steering committee up to five years following publication. A data sharing agreement (DTA) will need to be concluded between the representatives of the requesting institution and the University of Cape Town Lung Institute. Data sharing requests should be directed to keertan.dheda@uct.ac.za. Table S6 provides the accession codes for the WGS of the SARS-CoV-2 variants that could be sequenced for this study. The raw reads and raw count file for the RNAseq experiment has been deposited on the GEO website under the accession number GSE252508.

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

## Acknowledgements

This work was funded by the South African Medical Research Council (Covid Variant Consortium Grant) and the Center for Emerging and Neglected Diseases (Covid Catalyst Award to JM). K.D.'s lab also acknowledges funding from the European and Developing Countries Clinical Trials Partnership (grant nos. TMA-2015SF-1043 and TMA-1051-TESAII), UK Medical Research Council (grant no. MR/S03563X/1) and the Wellcome Trust (grant no. MR/S027777/1). We are indebted to the participants who took part in this study. We thank the Health Directorate of the City of Cape Town for providing access to appropriate health care facilities.

## Author contributions

Conceptualization: K.D., S.J. Laboratory Methodology: K.D., S.J., M.T., S.M., A.P., A.K., L.W., C.v.d.M., A.R., M.D., T.S., R.J., M.S., W.P., C.W., P.M., A. Sigal, J.L., and J.M. Clinical Investigation: K.D., A.E., S.O., T.P., A. Scott, A.G., E.M., and G.G. Funding acquisition: K.D. and S.J. Project administration: K.D. and S.J. Supervision: K.D., S.J., M.T., A.P. and A.E. Writing – original draft: K.D. and S.J. Writing – review & editing: K.D., S.J., S.M., M.S., W.P., A.G., E.M., P.M., J.L. and J.M.

## Competing interests

The authors declare no competing interests.

## Additional information

¹Division of Pulmonology, Department of Medicine, Centre for Lung Infection and Immunity, University of Cape Town Lung Institute, Cape Town, South Africa. ²Centre for the Study of Antimicrobial Resistance, South African Medical Research Council, Cape Town, South Africa. ³Department of Medical Biosciences, University of the Western Cape, Cape Town, South Africa. ⁴Division of Medical Virology, Wellcome Centre for Infectious Diseases in Africa, Institute of Infectious Disease and Molecular Medicine, University of Cape Town, Cape Town, South Africa. ⁵Division of Medical Virology, Faculty of Medicine and Health Sciences, University of Stellenbosch Tygerberg Campus; Medical Virology, National Health Laboratory Service Tygerberg, Parow, Cape Town, South Africa. ⁶Centre for the AIDS Programme of Research in South Africa (CAPRISA), Durban, South Africa. ⁷National Health Laboratory Service (NHLS), Cape Town, South Africa. ⁸HIV and Other Infectious Diseases Research Unit, South African Medical Research Council, Pretoria, South Africa. ⁹Department of Paediatrics and Child Health, University of Pretoria, Pretoria, South Africa. ¹⁰Department of Immunology, Faculty of Health Sciences, University of the Witwatersrand, Johannesburg, South Africa. ¹¹National Health Laboratory Services, Johannesburg, South Africa. ¹²Division of Immunology, Department of Pathology, Faculty of Health Sciences, University of Cape Town, Cape Town, South Africa. ¹³South African Medical Research Council, Cape Town, South Africa. ¹⁴National Institute for Communicable Diseases of the National Health Laboratory Service, Johannesburg, South Africa. ¹⁵SA MRC Antibody Immunity Research Unit, School of Pathology, Faculty of Health Sciences, University of the Witwatersrand, Johannesburg, South Africa. ¹⁶Africa Health Research Institute,

Durban, South Africa. [17]School of Laboratory Medicine and Medical Sciences, University of KwaZulu-Natal, Durban, South Africa. [18]Max Planck Institute for Infection Biology, Berlin, Germany. [19]Division of Pulmonary and Critical Care Medicine, Zuckerberg San Francisco General Hospital and Trauma Centre, University of California, San Francisco, San Francisco, CA, USA. [20]Department of Infection Biology, Faculty of Infectious and Tropical Diseases, London School of Hygiene and Tropical Medicine, London, UK. ✉e-mail: keertan.dheda@uct.ac.za

