## [Peer Review File · Nature Communications]

Frequency, kinetics and determinants of viable SARS-CoV-2 in bioaerosols from ambulatory COVID-19 patients infected with the β , δ or \omicron variantsEditorial Note: This manuscript has been previously reviewed at another journal that is not operating a transparent peer review scheme. This document only contains reviewer comments and rebuttal letters for versions considered at Nature Communications.

Reviewers' Comments:

Reviewer #2:

Remarks to the Author:

The authors have fully addressed my concerns, and the manuscript is a good fit for publication in Nature Communications.

Reviewer #3:

Remarks to the Author:

I have previously reviewed an earlier version of this manuscript submitted to another Nature journal. The authors have adequately addressed all my concerns.

Two minor comments:

Lines 175-176: Would strike phrase "the only ones that were specific and reliably ascertainable in contrast to non-specific or no symptoms"

Lines 262-265: I'm not sure what these sentences mean: "Notably, we not unequivocally demonstrate transmission but this is confounded by asymptomatic persons spreading COVID-19. There is no precise definition of what constitutes a 'super-spreader' in terms of aerosol culture positivity the term is not specific for infectiousness or transmission or both."

Reviewer #4:

Remarks to the Author:

I was asked to review the authors' responses to the original reviewer 1. I thank the authors for their efforts to address the concerns raised.

(1) "Firstly, I am missing a description of the cases in the results. Although the procedure is described in the methods, these studies require a clear description of cases in the results, including how people were selected or rejected."

Response: A section with the sub-heading 'Study design and description of cases' has now been added at the start of the result section.

Response 2: Appreciated. Is any information on past exposures or duration since vaccination available for the participants? I would advocate the use of "sex" instead of "gender" for the sake of this study, considering that the authors most likely assume it will be the biological, rather than the sociological, that impacts airborne transmission.

(2) "I disagree that this is evidence for the superspreader hypothesis. It is certainly interesting, and it could contribute to superspreading, as people who are positive for a long time could infect more people, but it is not evidence of superspreading, as one would have to know how many subsequent people were infected. It does show there is great variation in SARS-CoV-2 in humans."

We agree with the reviewer that there is no evidence of 'super-spreading' as such because we did not prove transmission. However, our data clearly demonstrates heterogeneity in the infectiousness of individuals with COVID-19. There have been a number of circumstantial lines of evidence to support this, but to our knowledge this is the first comprehensive clinically orientated study, showing that humans can produce infectious aerosols of COVID-19. In a nutshell, just under 30% were able to produce culture positive aerosols of $<5 \mu\text{m}$ at ~ 6 days after symptom onset. Thus, $<30\%$ of COVID-19 positive individuals, in the acute phase of the disease, demonstrated the ability to produce potentially infectious aerosols. This is exactly what the super-spreader hypothesis is broadly about: it follows the Pareto principle of the 80/20 or 70/30 rule, i.e. a small number of individuals, although having the respiratory infectious disease, will contribute to most of the transmission. The concept was further outlined in a paper by Lloyd-Smith et al Nature (Reference #18 in our paper) providing evidence that respiratory infectious diseases like SARS-CoV-1 are underpinned by the 'super-spreader' phenomenon.

In any event, this is not the primary hypothesis of the paper. The super-spreader concept is tangential.

Nevertheless, we do agree that we do not demonstrate spreading (proxy for transmission from one person to another). We have now made changes in the manuscript to clarify this aspect further and made it very clear that we did not demonstrate transmission. Furthermore, we have toned down and gone further by removing the words 'super-spreader' from the abstract, introduction, methods and results, and covered the above points, including the equipoise, in the discussion.

Some further thoughts on this aspect:

The case for SARS-CoV-2 transmission following the 80/20 Pareto principle was made during the early phase of the epidemic by Blasius [see Power-law distribution in the number of confirmed COVID-19 cases. *Chaos*. 2020; 30(9): 093123. PMID: 33003939].

A referenced study in this manuscript (Reference #24) looking at transmission in Hong Kong supports these figures, while another looking at transmission in Shenzhen, China, pegs the numbers closer to 80/10 [Bi et al. *Epidemiology and transmission of COVID-19 in 391 cases and 1286 of their close contacts in Shenzhen, China: a retrospective cohort study*. *Lancet Infect Dis*. 2020 Aug; 20(8): 911–919. PMID: 32353347]. Thus, there is prior art in this area.

We also listed as part of the limitations that "whilst we are able to potentially infer infectiousness of COVID-19 positive, we cannot make any conclusions about transmission as close contacts of study participants were not evaluated."

Nevertheless, to add clarity the following statement was added to the limitations section in the discussion section: "Finally, whilst we are able to potentially infer infectiousness of COVID-19 positive persons, we cannot make any conclusions about transmission as close contacts of study participants were not evaluated (thus we did not look at epidemiological evidence of onward transmission mainly because this is heavily confounded by transmission by asymptomatic persons)."

Response 2: This is sufficient.

(3) "Why is day 8 chosen as a cut off for figure 2D?"

Day 8 was selected as a cut-off based on previous reports (multiple independent studies) that SARS-CoV-2 could only be cultured from most nasopharyngeal or sputum samples up to 8/9 days after symptom onset [Jefferson et al. *Viral Cultures for Coronavirus Disease 2019 Infectivity Assessment: A Systematic Review* *Clin Infect Dis*. 2021;73(11):e3884-e3899. PMID: 33270107]. What is new in our report is that we detail in a comprehensive way the proportion that can produce culture positive aerosols.

Response 2: Thank you for the clarification.

(4) "Why are visits aggregated for some analyses? This does not make sense, as they were done during different times of the disease progression. Especially noted in figure 3A and table 1."

We think the aggregation in both inserts make sense.

Figure 3(A) is referring to the variant status. It is well known that variants may overlap over a period of time and there is not a sudden switch (over a period of a day or two) to a new variant. Thus, in this case it makes sense to aggregate the visits. We have also looked at the individual visits and we could not detect more than one variant in an individual over a period of time. There would be no need to show such a detailed analysis.

Regarding Table 1: The main hypothesis of the paper centres around the culturability of cough aerosols. Thus, Table 1 focuses on aerosol culture negative versus culture positive participants. The reason that Table 1 is further divided into aerosols of different sizes is that there is considerable controversy about the definition of aerosols, i.e. <5 micrometres versus <10 micrometres (as we have explained in the paper).

Table 1 could have been presented in many different ways, for example, splitting by symptoms status, vaccination status, etc. However, the convention is to group demographic and clinical characteristics that speak best to the primary hypothesis. We have in the online supplement, however, provided the analyses in different ways. We are not sure if the reviewer has looked at this?

Response 2: I do agree with the authors. I do struggle somewhat with some of the percentages in the tables when visits are cumulatively counted. I assume the status of the patient is then counted twice, correct? Why are the ORs not presented for all conditions? I may have missed this explanation if it was provided in the manuscript.

(5) "How does your data support epi data that vaccination is associated with reduced transmission?" We cannot draw any conclusions from our data regarding supporting the epidemiological data regarding vaccination. The reason is because of the small number of individuals who were vaccinated at the time for obvious reasons. We have made this clear in the paper.

The following paragraph in the discussion clarifies this point: "This is likely a consequence of the limited sample numbers in each sub-group. Reported epidemiological data indicate that vaccination is associated with reduced transmission of SARS-CoV-2 (albeit less effectively with the Omicron variant)²⁷; again, our sample size was inadequate to definitively address the question of whether, and to what extent, vaccination limits aerosolization capability. Although a higher proportion of culture negatives were vaccinated, we were not powered to detect small differences in effect size. Similarly, our study was not powered to identify genetic variants more likely to be associated with infectiousness."

Response 2: I find this sufficient.

(6) "It is stated that you have a higher portion of culture negatives in the vaccinated group, please graph this out, even if not statistically significant."

We did not understand this request. The proportions of individuals are clearly presented. We do not understand what the reviewer is asking for? What exactly does this reviewer want us to graph out? Furthermore, due to the timeline of vaccine roll-out, vaccination was disproportionately higher among Omicron cases (6/7) when compared to Beta and Delta cases (7/37). We therefore believe that we cannot reliably ascertain the impact of vaccination on culture status with the inability to control for variant status (as we have outlined in the previous response).

Response 2: I believe the original reviewer would have liked to see a graphical presentation of the data found in the tables to show, across all variant groups, the differences between vaccinated and unvaccinated. I find this, in principle, quite challenging as we do not know what history of previous exposures the participants have, making the situation very murky. It would be nice to know if the neutralizing antibody levels were correlated to vaccination status. I agree with the authors that their sample size does not allow us to infer this topic.

(7) "Can you comment on how long the simple 3-transcript blood-based biosignature can be detected? What are issues you may run into? When in the disease progression would this need to be done?"

This is a cross sectional study with prospective follow-up. However, we only collected and analysed

PAXgene tubes over a 48-hour period. The reviewer may be confused about this point.

One would need sampling at multiple timepoints, over 7 to 10 days in order to detect how long the biosignature could be detected. This is something that we are planning to do in the follow-on study. Nevertheless, to clarify this point we have made changes to the legend of Figure 1C and the methods section as follows: "Venous blood to evaluate host transcriptomics and soluble biomarkers such as neutralizing antibodies were collected over a 48-hour period (visit 1 and 2)." "Blood RNA sequencing was performed on 33 individuals (21 CASS-positive and 12 CASS-negative) using blood collected over a 48-hour period (visit 1 and 2; figure 1A)."

Response 2: The clarifications are appreciated. I understand that the original reviewer would have wanted this to be spelled out in the discussion.

(8) "The authors have a paragraph in the discussion stating that the higher Omicron transmission may be due to cell host factors, but they have already stated that cannot find differences between variants due to small numbers. You can then not also claim that there is no difference in ability to aerosolize." These aspects of the discussion, are speculative and argumentative, and nowhere have we made specific 'claims' as such. Yes, we are saying that differences would not be meaningful because of the small numbers in sub-groups.

The words 'may be' are used in the relevant paragraph thus toning down.

Response 2: This is fine.

(9) Other minor comments

Response 2: All addressed, thank you.

Reviewer 2

The authors have fully addressed my concerns, and the manuscript is a good fit for publication in Nature Communications.

Reviewer 3

I have previously reviewed an earlier version of this manuscript submitted to another Nature journal. The authors have adequately addressed all my concerns.

Two minor comments:

Lines 175-176: Would strike phrase “the only ones that were specific and reliably ascertainable in contrast to non-specific or no symptoms”

Response: The phrase has now been removed.

Lines 262-265: I’m not sure what these sentences mean: “Notably, we not unequivocally demonstrate transmission but this is confounded by asymptomatic persons spreading COVID-19. There is no precise definition of what constitutes a ‘super-spreader’ in terms of aerosol culture positivity the term is not specific for infectiousness or transmission or both.”

Response: The sentence was missing a word, and this is now clearer: “Notably, we DID not unequivocally demonstrate transmission but this is confounded by asymptomatic persons spreading COVID-19. There is no precise definition of what constitutes a ‘super-spreader’ in terms of aerosol culture positivity the term is not specific for infectiousness or transmission or both.”

Reviewer 4

I was asked to review the authors’ responses to the original reviewer 1. I thank the authors for their efforts to address the concerns raised.

(1) “Firstly, I am missing a description of the cases in the results. Although the procedure is described in the methods, these studies require a clear description of cases in the results, including how people were selected or rejected.”.

Response: A section with the sub-heading ‘Study design and description of cases’ has now been added at the start of the result section.

Response 2: Appreciated. Is any information on past exposures or duration since vaccination available for the participants? I would advocate the use of "sex" instead of "gender" for the sake of this study, considering that the authors most likely assume it will be the biological, rather than the sociological, that impacts airborne transmission.

Response: None of the participants had a documented history of COVID-19 prior to baseline diagnosis and enrolment into the study. The time from vaccination at enrolment, as well as the type of vaccine administered are known. However, considering the lack of any significant

association with vaccination (as imputed by low number), these details were not included in this study. Sex was used in place of gender.

(4) “Why are visits aggregated for some analyses? This does not make sense, as they were done during different times of the disease progression. Especially noted in figure 3A and table 1.”

We think the aggregation in both inserts make sense.

Figure 3(A) is referring to the variant status. It is well known that variants may overlap over a period of time and there is not a sudden switch (over a period of a day or two) to a new variant. Thus, in this case it makes sense to aggregate the visits. We have also looked at the individual visits and we could not detect more than one variant in an individual over a period of time. There would be no need to show such a detailed analysis.

Regarding Table 1: The main hypothesis of the paper centres around the culturability of cough aerosols. Thus, Table 1 focuses on aerosol culture negative versus culture positive participants.

The reason that Table 1 is further divided into aerosols of different sizes is that there is considerable controversy about the definition of aerosols, i.e. <5 micrometres versus <10 micrometres (as we have explained in the paper).

Table 1 could have been presented in many different ways, for example, splitting by symptoms status, vaccination status, etc. However, the convention is to group demographic and clinical characteristics that speak best to the primary hypothesis. We have in the online supplement, however, provided the analyses in different ways. We are not sure if the reviewer has looked at this?

Response 2: I do agree with the authors. I do struggle somewhat with some of the percentages in the tables when visits are cumulatively counted. I assume the status of the patient is then counted twice, correct? Why are the ORs not presented for all conditions? I may have missed this explanation if it was provided in the manuscript.

Response: Reviewer 2 is right, the status of the patient is counted twice as we considered them as two separate episodes, independent of each other during the natural history of the acute stage of infection. The ORs were not included in Table 1, since it is merely a description of the different participants across the defined study groupings. ORs have however been included in Table 2 where the outcomes of interest are displayed (culture status).

(6) “It is stated that you have a higher portion of culture negatives in the vaccinated group, please graph this out, even if not statistically significant.”

We did not understand this request. The proportions of individuals are clearly presented. We do not understand what the reviewer is asking for? What exactly does this reviewer want us to graph out? Furthermore, due to the timeline of vaccine roll-out, vaccination was disproportionately higher among Omicron cases (6/7) when compared to Beta and Delta cases (7/37). We therefore believe that we cannot reliably ascertain the impact of vaccination on culture status with the inability to control for variant status (as we have outlined in the previous response).

Response 2: I believe the original reviewer would have liked to see a graphical presentation of the data found in the tables to show, across all variant groups, the differences between vaccinated and unvaccinated. I find this, in principle, quite challenging as we do not know what history of previous exposures the participants have, making the situation very murky. It would be nice to know if the neutralizing antibody levels were correlated to vaccination status. I agree with the authors that their sample size does not allow us to infer this topic.

Response: No statistically significant association was found between neutralizing antibody levels and vaccination status. From the detailed histories taken none of the participants had an illness of previous COVID-19.